# Management of Upper Tract Urothelial Carcinoma in a Double Collecting System Kidney

**DOI:** 10.3390/jpm14020158

**Published:** 2024-01-30

**Authors:** Yarden Zohar, Bezalel Sivan, Ishai Mintz, Ben Hefer, Keren Rouvinov, Noa Shani Shrem, Nicola J. Mabjeesh

**Affiliations:** 1Department of Urology, Soroka University Medical Center, Faculty of Health Science, Ben-Gurion University of Negev, P.O. Box 151, Be’er Sheva 84101, Israel; bezalels@clalit.org.il (B.S.); yishaimi@clalit.org.il (I.M.); ben.hefer@clalit.org.il (B.H.); nicolam@clalit.org.il (N.J.M.); 2The Legacy Heritage Oncology Center, Dr. Larry Norton Institute, Soroka University Medical Center, Faculty of Health Science, Ben-Gurion University of the Negev, P.O. Box 151, Be’er Sheva 84101, Israel; kerenro@clalit.org.il (K.R.); noashs@clalit.org.il (N.S.S.)

**Keywords:** upper tract urothelial carcinoma, duplex collecting system, double collecting system, hemi-nephroureterectomy

## Abstract

Upper tract urothelial carcinoma (UTUC) in a duplex collecting system (DCS) is a relatively uncommon presentation with unclear management guidelines. Herein, we retrospectively reviewed all published cases of DCS with UTUC aiming to suggest personalized clinical care options for future cases. We conducted a systematic search for all cases of UTUC in DCS from published literature using the following keywords: UTUC, urothelial carcinoma (UC), collecting duct carcinoma, and DCS. The cases were summarized based on demographics, clinical presentation, predisposing risk factors, tumor location, management, and follow-up. We present an additional case based on our experience with a 69-year-old female with high-grade (HG) UTUC of the upper moiety in complete DCS. The patient underwent a robotic upper pole hemi-nephroureterectomy (hemi-NU) with a common sheath distal ureterectomy and a bladder cuff, followed by lower pole ureteral reimplantation. Overall, 34 patients with 35 renal units of UTUC in DCS were included and analyzed. To conclude, UTUC of DCS is rare and underreported. Hence, it is difficult to define a standard treatment. Although hemi-NU has been previously described, to the best of our knowledge, this is the first case report of robot-assisted hemi-NU for complete DCS with single-moiety UC.

## 1. Introduction

Urothelial carcinoma (UC) is the fourth most common solid organ malignancy; however, only 5–10% arises from the upper urinary tract [1,2,3]. Thus, the prevalence of primary upper tract urothelial carcinoma (UTUC) in duplex collecting system (DCS) is underreported with an unknown estimation rate.

DCS is one of the most common congenital anomalies of the urinary tract (UT), accounting for 0.7–1% of all anomalies [4]. It is characterized by a duplex kidney, where every unit acts as a separate pelvicalyceal system that finally drains into the bladder via either a complete or an incomplete collecting system. The collecting system is classified according to the level of ureteral fusion: (1) proximal fusion at the level of the ureteropelvic junction (UPJ) is associated with a duplex kidney and single ureter; (2) an incomplete system is defined as a bifid ureter that fuses proximal to the ureteral orifice; and (3) a complete system occurs when the ureters drain the bladder separately following the Weigert–Meyer rule [5]. DCS can then be further complicated by additional congenital anomalies such as obstruction, reflux, ectopic ureter, or ureterocele. Ureterocele is an intravesical cystic dilatation that appears with DCS in 80% of the cases. It is six times more common in females, and when revealed in older age, it is most likely to be an orthotopic protrusion found at the level of the vesicoureteral junction [6].

The association between congenital anomalies of the UT and increased risk of malignant neoplasms remains under investigation. A historical population-based cohort study followed up with army recruits with congenital anomalies of the kidney and UT for over three decades to estimate the hazard ratio for tumors. The study concluded a nine times higher risk for malignant neoplasms in females younger than 45 years of age; however, the opposite was observed in the male population, where the risk increased over the age of 45 years [7].

Considering its rarity, there is no consensus regarding the management of UTUC in DCS. Herein, we summarize the available information regarding UC in DCS and present our experience to define a particular, acceptable way of patient care in the robotic era.

## 2. Materials and Methods

### 2.1. Research Strategy

The research strategy was systematic and included all UC cases in the DCS. A preliminary search of published scholarly literature was conducted in PubMed (https://pubmed.ncbi.nlm.nih.gov/) on 1 September 2023, Mendeley library (https://www.mendeley.com/search/) on 1 September 2023, and Google Scholar on 1 September 2023 using the following keywords: upper tract urothelial carcinoma (UTUC), urothelial carcinoma (UC), collecting duct carcinoma, duplex collecting system (DCS), double collecting system, hemi-NU, partial moiety nephrectomy, and nephroureterectomy (NU) for double collecting system carcinoma. 

### 2.2. Study Selection and Data Extraction

Following data collection, manual filtration was performed based on the following exclusion criteria.

(1)Any case of UT malignancy out of urothelial origin was excluded (i.e., renal cell carcinoma).(2)Cases of urothelial malignancy of the UT in a kidney with anomalies other than DCS were excluded.(3)Any case of hemi-NU due to a nonfunctioning moiety was excluded.  Articles were selected based on the following inclusion criteria:(1)Any original case report or series of UTUC in a DCS kidney was included.(2)Cases of urothelial neoplasia other than carcinoma in DCS or special UC variants (sarcomas) were included.(3)Cases of UTUC in DCS that were not surgically managed were included.(4)Abstracts and publications without full text were included.(5)A single case of a paper published with English abstract and Japanese content was translated via Google Translate for specific clarifications. Most of the information retrieved was in English; therefore, it has been included.

### 2.3. Data Analysis

The selected cases were manually analyzed, and the extracted data were summarized in a table based on the available information. Subsequently, an original case report was added, drawing from our clinical experience. Demographic information, radiological findings, clinical management, surgical and follow-up data were also included. A detailed summary of all cases is presented in Table 1. Descriptive data based on 34 patients, encompassing 35 units of UTUC in DCS, are summarized in Table 2.

## 3. Case Presentation

A 69-year-old female patient with intermittent hematuria that persisted for 18 months presented to the emergency department with acute general weakness. The blood hemoglobin (Hb) level was 8 g/dL and the creatinine (Cr) level was 0.6 mg/dL. Initial workup included blood transfusion and computed tomography urography (CTU) scan. CTU revealed complete DCS of the left kidney with a 4 cm space-occupying mass in the renal pelvis of the upper moiety (Figure 1). There was no evidence of distal or local metastatic disease on the chest–abdomen–pelvis CT scan. A tissue biopsy was performed during diagnostic ureteroscopy (URS), conforming T1 high-grade (HG) UC of the renal pelvis of the upper moiety. No endoscopically observed involvement of the lower moiety or either of the ureters was observed. Notably, the upper ureteral orifice was identified within a small ureterocele (Figure 2).

### 3.1. Management

Following the diagnosis, definitive surgical therapy was planned for the cT1N0M0 HG UC. Although NU is the standard of care for HG UTUC, there is no consensus regarding the recommended procedure for UTUC in DCS. As the medical workup confirmed a localized tumor in the upper pole renal pelvis, this moiety was treated as an independent pelvicalyceal functional unit for risk stratification. In accordance with the European Association of Urology (EAU) guidelines on UTUC, kidney-sparing surgery is recommended, taking into consideration case-specific factors and after a comprehensive patient consultation [34]. Consequently, the patient was presented with the option of nephron-sparing hemi-NU. After obtaining the patient’s consent, she underwent robot-assisted laparoscopic upper pole hemi-NU with distal common sheath ureterectomy, including a bladder cuff containing the ureterocele and reimplantation of the lower moiety ureter (Figure 3).

### 3.2. Surgical Intervention

Prior to robotic arm docking, a revised cysto-ureteroscopy confirmed that there was no evidence of a macroscopically visible tumor in the lower moiety pelvis and ureter. The Da Vinci Xi robotic system was used with the introduction of three ports inserted in a right paramedian line, with an additional 12 mm umbilical port inserted for the assistant. Initially, robotic targeting was programmed towards the hilus of the left kidney. After exposure of the retroperitoneal space, the upper moiety ureter was identified and rostral dissection was performed as far as the upper moiety was made. At that point, the renal vessels were identified as the main vein and artery, which ran towards the lower moiety and then bifurcated to provide blood supply to the upper moiety. To preserve the lower moiety vasculature, indocyanine green (ICG) dye was used simultaneously with upper pole artery clamping. At this point, a visual demarcation line could be identified between the ischemic upper pole moiety and vital lower pole moiety. After the upper pole vasculature was ligated using a robotic vascular stapler, the upper pole was carefully separated and a double-layer suture line was made for the remaining lower moiety as acceptable for classic partial nephrectomy, followed by arterial unclamping. The total warm ischemic time was 15 min. After the dissection and separation of the upper pole kidney and proximal ureter (passing underneath the blood vessels of the lower pole), robotic retargeting was performed, focusing on the urinary bladder. The ureters were then dissected caudally towards the bladder as far as the point where the two ureters were enveloped within a common sheath, piercing the bladder wall. In the urinary bladder, the ureters were dissected together off the wall with a bladder cuff, including the ureterocele. The lower pole of the ureter was preserved and separated 1 cm above the common sheath. To prevent local neoplastic seeding, prior to reimplantation, the upper pole with its clipped ureter was carefully pulled from the passage between the blood vessels and the lower pole pelvis and placed in a laparoscopic bag. The remaining ureteral stump of the lower moiety was then re-implanted into the bladder wall over a 6FR/26cm double-J stent, and a Jackson–Pratt (JP) drain was placed adjacent to the anastomosis.

### 3.3. Operative and Postoperative Course

The overall surgery time was 220 min, including the pre-robotic cysto-ureteroscopy time, with an estimated blood loss of 400 CC. The Hb level dropped from 12.4 to 10 g/dL; however, the patient was hemodynamically stable, and no blood transfusions were administered intraoperatively or postoperatively. The postoperative course was uneventful, with a Clavien–Dindo classification system score of 0. Cr levels remained unchanged at 0.8 mg/dL. The patient initiated a diet and mobility plan on postoperative day one, the urinary catheter was removed on postoperative day two, and the drain was withdrawn the following day. Discharge occurred on postoperative day three.

A week later, the patient returned to the emergency department experiencing weakness, vomiting, and diarrhea. The physical examination findings were unremarkable. Blood work results indicated a slight elevation of white blood cells (WBCs) to 12,000, and C-reactive protein (CRP) to 5 (upper normal limit defined until 5). Urine and blood cultures were negative. Nevertheless, the patient’s condition improved with empirical antibiotic therapy. Subsequently, the patient was discharged with a recommendation for a complete 7-day course of oral ciprofloxacin therapy.

### 3.4. Follow-Up

Six weeks post-surgery, the patient was followed up in an outpatient clinic. Blood tests confirmed a Cr level of 1.0 mg/dL, indicating satisfactory renal function. The patient exhibited a good performance status. The histopathological report confirmed mucinous HG urothelial carcinoma with the largest dimension measuring 2 cm. Mucicarmine staining was positive. The tumor focally invaded the kidney; however, the surgical margins were negative for malignancy. The common sheath ureters were tumor-free. The disease staging, based on the pathology report, concluded T3NxM0 urothelial carcinoma [35].

The case was presented at our institutional multidisciplinary tumor board, where the decision for adjuvant chemotherapy treatment (ACT) with cisplatin and gemcitabine protocol was made. Before ACT, a positron emission tomography scan (PET-CT) was performed, excluding any distant or local metastases. Notably, the patient’s Cr level remained stable for six months after the procedure, despite ACT (0.95–1 mg/dL). Two months later, the patient underwent a cystoscopy. The DJ stent was removed, and the bladder cystoscopy revealed no evidence of a tumor. At the 6-month follow-up, the patient underwent CTU, demonstrating preserved corticomedullary and excretory phases of the left lower moiety pelvicalyceal unit, with only local postoperative changes. No evidence of local recurrence was observed. At the 9-month follow-up, the patient underwent diagnostic URS and cystoscopy, with no evidence of disease (NED). Subsequently, she underwent regular follow-ups every three months with a urologist, concurrently maintaining active oncology surveillance. It is noteworthy that during this period, the patient reported fatigue and nausea, which subsequently resolved upon the completion of ACT. Her current physical status is unremarkable, with a score of 0 in the ECOG score.

## 4. Results

Overall, the data analysis encompassed 34 cases derived from a retrospective systematic search and case presentation. Among these, two cases were bilateral; one presented bilateral DCS with bilateral UTUC, while the second case exhibited bilateral tumors, with only one side having the anatomical variation in DCS (Table 1: patients 17 and 13, respectively). Therefore, the corrected total number of UTUCs in the duplex system was 35. The mean age was 65.8 ± 11.1 ranging from 52 to 87, with 21 males and 13 females. Only 11 cases reported Cr levels, of which 3 had values higher than normal. All patients, except for one, presented with intermittent hematuria, which was the cardinal symptom and most commonly the only symptom. Notably, 61.7% of cases were reported in Eastern Asia. Regarding anatomy, 51.42% of UTUC cases had an incomplete system and 36.42% had complete DCS. The side of the involved kidney was not statistically significant. Tumor location varied, and there was no correlation between tumor location and whether duplicated system anatomy was complete or incomplete. In fact, 50% of tumors involved the upper moiety of the renal pelvis and proximal ureter. Out of 18 incomplete collecting systems, 8 tumors developed at the level of bifurcation. Among the 34 cases, 28 were managed surgically; however, only 5 cases were managed with hemi-NU. Our presented case was the only one managed solely using a minimally invasive approach. Out of the total 34 cases, only 18 provided long-term follow-up data; 72% (13/18) of these cases showed no evidence of disease (NED) after 6 months of follow-up, and 38% (7/18) maintained NED for 2 years. Regarding post-hemi-NU follow-up, three out of five (60%) patients reported NED after 6 months, of whom two survived over 2 years with no recurrence. The data for the remaining two cases were missing. Table 2 presents the characteristics of the population in this study.

## 5. Discussion

The present study consolidates all available published data on UTUC in DCS and introduces an uncommon surgical management approach based on our experience. Although relatively rare, hemi-NU for single-moiety UTUC in duplex systems has been described in the past. However, few of these surgeries were primarily for diagnostic purposes rather than definitive treatment, at times when diagnostic URS was unavailable. Furthermore, these procedures were typically performed through an open approach and were reserved for selected cases where radical NU would have led to irreversible kidney failure or death [10,11,13,16,21,25,28]. To the best of our knowledge, this is the first published robotic hemi-NU aimed at preserving renal function in a patient for whom NU was not an absolute contraindication.

### 5.1. Hemi-Nephroureterectomy Considerations

As previously mentioned, open radical NU with a bladder cuff has been the gold standard treatment for HG UTUC over the last five decades [34,36]. To highlight the commitment to this traditional approach, Chen et al. reported a case of bilateral upper moiety UTUC in a patient with bilateral DCS, which ended with bilateral radical resection and lifelong hemodialysis without considering a nephron-sparing procedure [8].

Technological advancements have facilitated minimally invasive nephron-sparing procedures, with increasing popularity. Numerous publications have focused on endoscopic management and distal ureterectomy for low-grade (LG) or morbid patients with HG UC, now supported by international guidelines [33,34,37]. Additionally, over the last decade, partial nephrectomy has become the gold standard of care for small renal masses, in contrast to the previously prevalent radical nephrectomy [38].

Hemi-NU is commonly discussed in the management of nonfunctioning moieties in pediatric patients with DCS. Currently, this procedure is frequently performed using a laparoscopic approach. While the open approach is also relatively common, a gradual shift towards the robotic technique is imminent [39]. In adults, hemi-NU is much less common. The literature review encompasses only selected cases, most of which are non-oncological procedures [40,41].

According to our data extraction, open hemi-NU for UTUC was first described in 1987 by Budd et al., who performed the procedure on a 40-year-old male patient with an upper pole UC of a complete DCS of the right kidney. An additional case was described in 1991 in Japan of an incomplete collecting system with UC. Follow-up was not available for these two cases, raising questions about the long-term outcomes of the procedure [8,13].

In 2019, Karray et al. reported a hemi-NU involving the resection of the upper pole moiety containing a suspected mass in a complete DCS of the right kidney. The upper pole ectopic ureter was inserted into the right prostate lobe. The presurgical evaluation revealed no evidence of any suspected lesion surrounding the periprostatic tissue and distal ureter; therefore, the prostate was preserved. Pathology revealed unifocal, infiltrative pT2G2 UC. Follow-up was recorded for 2 years, and notably, the surrounding peri-distal ureteral tissue in the prostate remained intact. At the diagnostic URS 2 years later, there was no stump recurrence [28].

Our data report five hemi-NU procedures out of eighteen surgeries for UTUC in a DCS, with four cases performed in a complete system and only one in an incomplete collecting system. Notably, tumor location varied between cases, and in all five cases, the pathology confirmed HG UC. Long-term follow-up was available for three of the five cases, and NED was confirmed after 6 months. All three NED cases were in the complete DCS group, but the missing long-term data regarding a single case of hemi-NU in an incomplete duplex system limits the comparison.

Theoretically, hemi-NU for complete DCS may be considered safer owing to its discrete, unconnected ureteral course and opening. Even in the presence of a common sheath, there is no direct intimal mucosal connection; therefore, the rate of direct contamination is relatively low compared with an incomplete system. Additionally, as noted in Table 2, tumor involvement at the bifurcation point in an incomplete collecting system is relatively common and requires radical management with complete system resection.

### 5.2. Surgical Management of UTUC in Malformed Kidney

Cullivan et al. recently reported a laparoscopic left hemi-NU in a horseshoe kidney with UTUC. This case is noteworthy because it strengthens the rationale beyond our therapeutic intervention, involving the dissection and separation of anatomically related renal parenchyma containing malignant tissue. However, while both procedures are defined as hemi-NU and include resection of a single functioning pelvicalyceal unit, there are notable differences. In DCS, the two ureters are anatomically connected either directly in an incomplete system or enveloped within a shared common sheath. An anatomically closely related collecting system with multiple possible connection points presents surgical challenges; more importantly, it might potentially be a point of recurrence.

In contrast, the hemi-NU of a horseshoe kidney is more comparable to a single kidney radical NU, as the bilateral collecting systems are completely anatomically separated by the renal parenchymal isthmus [42]. In addition, the two moieties of a duplex system share common blood vessels, posing challenges in surgical intervention and resection with a higher chance of surgical failure.

Crossed fused renal ectopia (CFRE) is a rare congenital anomaly in which the two kidneys are fused together and lie on the same side [43]. Only three cases of UTUC in CFRE have been described in the literature. All cases were managed surgically—two with open NU and one with laparoscopic NU. Comparing NU for CFRE with complete DCS hemi-NU, in both procedures careful management of the renal vessels is essential since the two pelvicalyceal units are usually supplied by a single aortic branch that then bifurcates [44].

### 5.3. Pathophysiology of UTUC in Malformed Kidney

Theoretically, it is believed that urine stasis caused by malformed collecting systems increases the risk of cancerous changes due to chronic inflammation [45]. However, in this case, squamous cell carcinoma (SCC) was the most likely histological variant [26,46]. Still, SCC accounts for less than 1% of all UTUC. Our results correlated with general statistics, with only 1 out of 34 cases being positive for SCC, raising doubts regarding the cancer secondary to the chronic inflammation hypothesis. In fact, for DCS, urine reflux and stasis are significant, especially when they co-exist with ureterocele; however, the majority of patients have UC rather than SCC. Notably, 4 out of 23 reported histological results had an aggressive variant of UC, consistent with the overall statistics of histopathological UTUC variants in non-malformed kidneys of 25% [1,47]. Therefore, overall, anatomical anomalies did not increase the risk of aggressive variants.

The principle of radical NU with a bladder cuff is to prevent intraluminal seeding by antegrade urine flow. In fact, 22–47% of patients with UTUC present with concomitant bladder UC at diagnosis, 20–50% of patients develop secondary bladder cancer following radical NU throughout their lifetime, and more than 60% of these patients will present with recurrence within 2 years postoperatively [48,49]. The incidence of bilateral UTUC is 17%, and the most important prognostic factors for recurrent contralateral UTUC are an estimated glomerular filtration rate (eGFR) of less than 30% ml/min and a primary multifocal ureteral tumor causing collecting system “seeding contamination” [50]. The seeding theory was enforced by late biological profiling research on UTUC tumors followed by bladder tumor recurrence. Research has revealed that both upper tract and bladder tumors have similar molecular features. For example, metachronous muscle-invasive urothelial carcinoma (MIUC) recurrence following UTUC (which overexpresses FGFR3) may have an overexpression of FGFR3 as well (compared to primary bladder MIUC) [51].

In addition to the “intraluminal seeding” hypothesis, the pathophysiology of UTUC is also affected by environmental exposure. The “field cancerization” is directly associated with UC and particularly UTUC. Carcinogens alter cellular microenvironments and promote mutations. The urinary system, which drains large amounts of carcinogens in the body, is highly vulnerable [52,53]. Data analysis showed that the duplex system UTUC is much more prevalent in East Asia. The reported rate of UTUC in Japan is 40%, compared to only 5% worldwide. Epidemiological studies have suggested that chronic arsenic exposure is the leading cause of the increased incidence. In Taiwan, over 100 million people are at risk of drinking arsenic (As)-contaminated water. In Japan, it is most prevalent in the Hokusetsu region [54,55]. Moreover, the consumption of Aristolochia-containing plants, which are common in this region (particularly in China), further increases the risk of UTUC [56].

### 5.4. Oncological Considerations

The oncological management of UTUC is intricate, aligning with the patient’s overall medical status, expectations, and tumor features to ensure better outcomes. For low-risk/LG disease, renal-sparing procedures are recommended. However, the management of HG/high-risk UC is more restrictive and requires careful multidisciplinary planning [34].

Although conclusive results have not been published, neoadjuvant chemotherapy (NAC) may be considered for patients with advanced disease, or where postoperative renal function precludes ACT. Studies have shown that NAC reduces disease recurrence and mortality rates. Furthermore, some research reports pathological downstaging in case of metastatic disease, leading to favorable operative outcomes [57,58]. The usage of immunotherapy as a neoadjuvant for cisplatin-unfit patients was assessed in a small phase II randomized control trial (RCT) and did not prove to be advantageous [59].

After NAC, surgery is contemplated. According to the EAU guidelines, kidney-sparing surgery for UTUC should be considered in cases of a single kidney or impaired renal function based on specific cases. In this instance, we were not obligated to preserve renal function since the patient’s GFR was >60 mL/min and she did not exhibit any medical comorbidities that could cause a decrease in renal function [34]. However, based on the *Nature* review from 2015, nowadays, thanks to precise surgical tools and sensitive radiological examinations, nephron-sparing procedures can be considered even in healthy young adults, if close clinical follow-up can be maintained [49]. Open radical NU (RNU) with bladder cuff excision is the gold standard therapy for HG/high-risk UTUC [34]. The minimally invasive approach is acceptable when oncological principles are maintained. Current data confirms that the robotic approach is superior to open surgery in terms of postoperative complications and not inferior in terms of oncological outcomes, thus increasing its popularity [60].

The primary concern with renal sparing surgical therapy, such as hemi-NU, for UTUC, lies in the oncological outcomes. Firstly, considering the relatively high rate of seeding contamination for UC, minimizing intraoperative contamination is crucial [53]. According to the revised and updated 2023 EAU guidelines, for UTUC, diagnostic URS can be employed, preferably without a biopsy, only when imaging and/or voided urine cytology has proven insufficient for the diagnosis and/or risk stratification of patients suspected to have UTUC [34].

Our analysis reveals that 15 out of 34 cases (44%) were diagnosed with a UC based on retrograde pyelography (RPG) or diagnostic URS. In DCS, especially in incomplete anatomy, the significance of intraluminal contamination becomes crucial when planning hemi-NU. However, the underreported follow-up information in Table 1 complicates the estimation of the rate of seeding contamination for DCS with UTUC diagnosed with RPG/URS. Nevertheless, our patient, who underwent diagnostic ureteroscopy (URS) with biopsy prior to radical resection, was reassessed by cysto-ureteroscopy and showed no evidence of recurrence. Still, in alignment with updated guidelines, consideration should be given to minimizing the use of these interventional diagnostic tools when non-interventional imaging is feasible [34].

When conducting radical resection, bladder cuff excision is imperative, significantly reducing the recurrence rate [61]. In contrast, intraoperative nodal management is not obligatory. In 30–40% of radical NU surgeries, a suspected lymph node is visible. However, the benefits of lymph node dissection (LND) for node-negative (cN0) UTUC have not yet been established [62]. Notably, while LND improves pathological staging, the updated EAU guidelines do not recommend LND for cT1N0 disease [34]. Consequently, LND in our patient was not performed. For cT2-T4, a recent multi-institutional study concluded that LND did not improve survival in patients with cN0 disease. In positive-node disease, prognosis is directly related to the number of positive nodes removed and the dissection template [62]. It is essential to align the primary tumor location to the nodal dissection template because ureteral tumors are locally more advanced and typically metastasize earlier than renal pelvic tumors [63].

Platinum-based ACT is recommended for pT2-T4/any N/M0 disease within 90 days of surgery to prolong disease-free survival (DFS) [64]. For cisplatin-ineligible patients, carboplatin can serve as an alternative [65]. Immunotherapy, particularly checkpoint inhibitors (CIs), may also be considered. These ICs are frequently used as an alternative treatment for platinum-resistant disease or as maintenance therapy after cisplatin/carboplatin + gemcitabine adjuvant chemotherapy in cases of non-progressive disease [66,67,68,69,70,71]. Current cancer research is focusing on immune-phenotype-based therapy to enhance tumor response [72]. Radiotherapy is less commonly employed, with limited information available regarding its efficacy as a combination therapy. Typically, it is used for targeted lesions [73,74].

### 5.5. Follow-Up Plan

Conservative surgical management necessitates close follow-up; therefore, developing a detailed follow-up plan becomes crucial. According to the EAU guidelines, URS is required after three months. However, in our patient, this was not feasible because of severe lymphopenia resulting from the chemotherapeutic regimen. To minimize the risk of infection, given the normal Cr levels, the patient was monitored with cystoscopy and CTU as long as the lymphocyte level remained low. As mentioned, after bone marrow recovery, at 9 months follow-up, the patient underwent diagnostic URS and cystoscopy.

While the need for a defined follow-up protocol persists, we relied on expert experience and the patient’s condition, formulating a tailored follow-up plan with 3-month intervals, alternating between the commonly used diagnostic tools: imaging, cytology, and endoscopy.

### 5.6. Prognosis

The primary prognostic factors for UTUC, irrespective of the collecting system anatomy, along with the patient’s characteristics are tumor stage and grade. The 5-year cancer-specific survival (CSS) in non-muscle-invasive UTUC is 86%, while for muscle-invasive disease, it is 70%. About 9% of patients present with metastatic disease, with a CSS of 39% [75,76]. Our patient was considered a suitable candidate for this surgery, benefiting from a favorable prognostic profile at the initial presentation. She was younger than 80 years, with no medical illnesses. Her BMI was 22, with an eGFR greater than 60 mL/min. She had no personal or familial history of malignancy and no previous exposure to predisposing environmental/occupational risk factors. The patient’s compliance with medical follow-up is unremarkable, allowing for the early detection of recurrence.

### 5.7. Limitations

Although the UTUC of DCS is uncommon, a defined management algorithm is necessary. This study serves as an initial basis for future management, providing a summary of all available data. However, our data processing method had a few limitations. The majority of the information was collected retrospectively, resulting in a lack of essential information. As mentioned, only 18 out of 34 patients reported a long-term follow-up. Moreover, the limited number of hemi-NU cases weakens the final conclusion due to low statistical power. Despite the promising results of our case, it is insufficient to be implicated in future clinical management. Nevertheless, we encourage transformation in clinical perspectives regarding the surgical approach in times of technological renewal.

## 6. Conclusions

In conclusion, UTUC of DCS has been previously documented; however, there is currently no consensus on the clinical management of such cases. In fact, this case serves as a pivotal milestone for the future management of localized aggressive tumors that can be completely resected with the advancement of surgical techniques in the robotic era. The integration of strict follow-up for the early detection of possible disease recurrence ultimately improves the patient’s quality of life and enables the safe administration of complementary adjuvant therapies. We advocate for the adoption of a conservative, organ-sparing surgical approach when both technology and expertise are readily available.

## Figures and Tables

**Figure 1 jpm-14-00158-f001:**
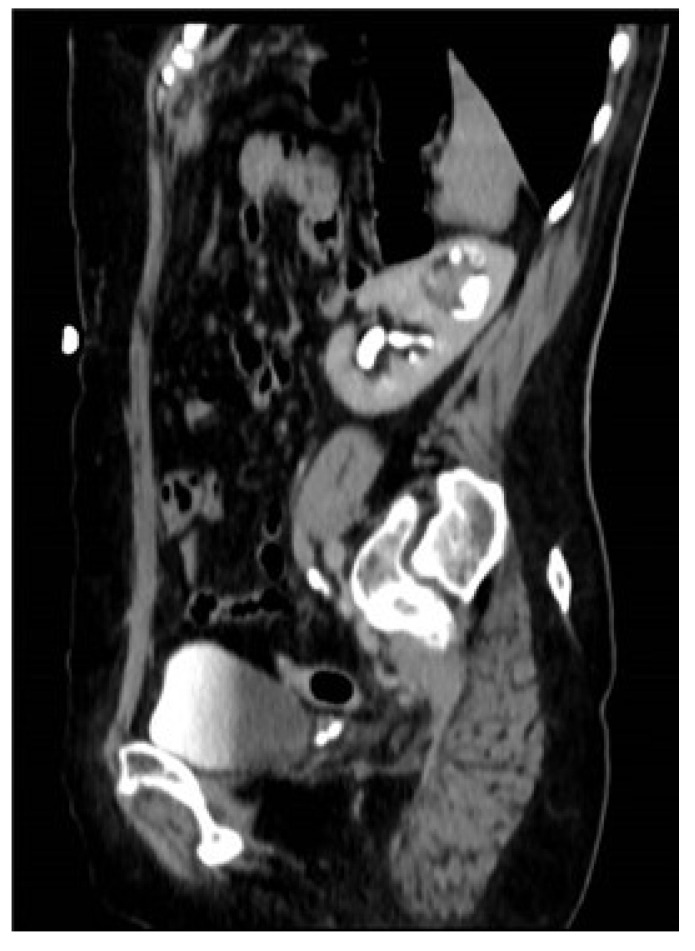
Sagittal section CT scan demonstrating the left kidney with a space-occupying lesion in the upper moiety.

**Figure 2 jpm-14-00158-f002:**
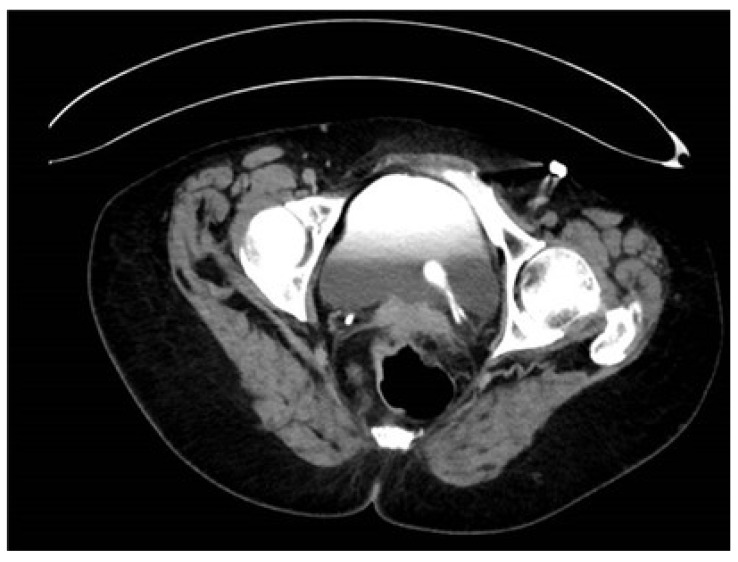
CT scan showing ureterocele of the upper moiety in the bladder.

**Figure 3 jpm-14-00158-f003:**
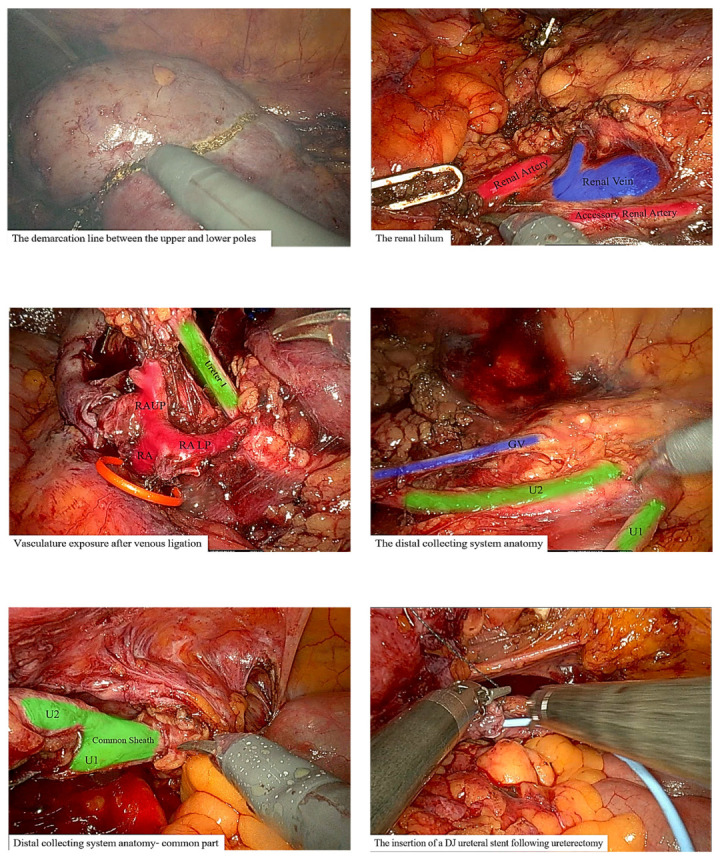
An intraoperative step-by-step demonstration as described in the surgical intervention Section 3.2. Red, arteries; blue, veins; green, collecting system including ureters and a common sheath. GV, gonadal vein; U1, ureter of upper moiety; U2, ureter of lower moiety; RA, renal artery; RAUP, renal artery of the upper pole; RALP, renal artery of the lower pole.

**Table 1 jpm-14-00158-t001:** Summary of all cases of upper tract urothelial carcinoma (UTUC) with duplex collecting system (DCS).

	Reference	Year	Region	Age	Gender	Presentation	Exposure	Cr	Diagnosis	Anatomy	Side of Duplicated System	Location of Tumor	Management	Approach	Pathology	Follow-Up
1	Kumon et al. [8]	1981	Japan	52	M	N/A	N/A	N/A	N/A	Incomplete	Right	At the junction site of the incomplete duplicated ureters	N/A	N/A	N/A	N/A
2				67	M					Incomplete	Left					
3				70	F					Incomplete	Right					
4				67	F					Incomplete	Left					
5				62	F					Incomplete	Right	Near the junction site				
6	Banya et al. [9]	1986	Japan	78	M	Hematuria	N/A	N/A	IVU, RPG	Incomplete	Right	Polyp-like filling defect in the lower segment of duplicated ureter at about 4 cm from the fusion of the ureters	Radical NU + partial cystectomy	N/A	Prior to NU, patient underwent diagnostic URS with tumor resection, primary pathology confirmed T1 LG disease. Histopathology after NU was negative for malignancy	N/A
7	Tudor et al. [10]	1986	Japan	56	F	Hematuria	N/A	N/A	IVU, US	Complete	Right	Right lower moiety ureter near the orifice	Local excision with ureteroureterostomy	Laparotomy	N/A	N/A
8	Budd et al. [11]	1987	United Kingdom	40	M	Hematuria	N/A	N/A	IVU	Complete	Right	Right upper moiety pelvis	Hemi-NU	Laparotomy	N/A	N/A
9	G. SREENEVASAN et al. [12]	1987	Malaysia	67	M	hematuria	N/A	N/A	IVU	Incomplete	Right	Lower moiety proximal ureter 2 cm above fusion	Radical NU + bladder cuff	Laparotomy	Papillary TCC	2 years follow-up NED
10	Gepi-attee [13]	1991	Japan	74	M	Hematuria	Smoking	N/A	US, RPG	Incomplete	Left	Left upper moiety	Hemi-NU	Laparotomy	G2pT2 invasive papillary UC resection margins free of tumor	N/A
11	Asase et al. [14]	1992	US	66	M	Hematuria, weakness	N/A	N/A	US, RPG	Incomplete	Left	Left renal pelvis of lower moiety	Radical NU with bladder cuff	Laparotomy	N/A	N/A
12	DUDAK et al. [15]	1995	US	81	M	Hematuria	No per anamnesis	1.2	IVU, CT, RPG	Complete	Left	Left upper moiety ectopic ureter distal part adjacent to orifice	Radical NU following nuclear renal scan (R 32% L 68% Cr level 1.2 mg/dL)	Laparotomy	pTa G I-II/III multifocal involvement in upper moiety pelvis	7 months NED
13	Tan et al. [16]	1996	Taiwan	62	M	Hematuria, dysuria, suprapubic pain, cachexia	Lives in Taiwan (Blackfoot disease endemic area, long-term exposure to inorganic arsenic)	10.7 mg/dL	US, CT, RPG	Incomplete	Left	Right distal ureter Left middle ureter area, after the point of ureteral fusion	Right—radical NU Left—segmental ureterectomy with bladder cuff and ileal ureter	Exploratory laparotomy	Right—pT1 G1 TCC Left-pT1G2 TCC	Follow-up RPG 6 months later- no recurrence, Postoperative Cr 4.9 mg/dL 2 years follow-up—NED
14	Tamada et al. [17]	1998	Japan	72	M	Hematuria	N/A	N/A	CTU, RPG	Complete	Left	Left upper moiety	Radical NU	N/A	TCC	N/A
15	KAWAMURA et al. [18]	1998	Japan	67	F	Hematuria	History of surgical and irradiation therapy for breast cancer	N/A	IVU, CT	Blind-ending bifid ureter	Right	The bifid ureter was filled with serous dark fluid with a narrow lumen	Radical NU	Laparotomy	pT3N0M0UC	Patient received adjuvant chemotherapy. 6 months follow-up NED
16	Chen et al. [8]	2002	Japan	66	M	Hematuria	No per anamnesis	1	IVU + RPG	Incomplete	Right	Right lower moiety	Radical NU	Laparoscopic	TCC in pelvis, ureters were free of tumor	6 months NED
17	Chen et al. [8]	2002	Japan	58	M	Hematuria	No per anamnesis	1.9	IVU, CT	Complete	Right	Renal pelvis of upper moiety	Bilateral radical NU + bladder cuff	Laparotomy	TCC	After 1 year follow-up, patient developed bladder tumor, he died of sepsis 2 years after NU
										Incomplete	Left	Mass in left lower moiety				
18	Chen et al. [8]	2002	Japan	65	F	Intermittent hematuria for 2 months followed by an acute presentation with syncope and N and V	No per anamnesis	16.6 mg/dL	CT (under hemodialysis), APG + RPG	Incomplete	Left	Distal end of the left lower moiety ureter immediately above the junction site	Radical NU + bladder cuff	Laparotomy	TCC	Maintenance hemodialysis 6-month follow-up cystoscopy revealed bladder recurrence. At 2-year follow-up, patient received intravesical chemotherapy
19	Takagi [19]	2002	Japan	66	M	Left flank pain	Lives in Japan	N/A	IVG, CT, MRI	Complete	Left	Papillary tumor from the left ureteral orifice of the lower renal moiety	Diagnostic URS confirmed pTaG2 UC followed by radical NU with partial cystectomy	Not mentioned	pT3G3N1MX UC	Patient received adjuvant chemotherapy (M-VAC) that was initially discontinued due to severe side effects, however, re-administrated 22 months later because of recurrence detected in retroperitoneal lymph nodes
20	Hisataki et al. [20]	2002	Japan	43	F	Hematuria and left flank pain	N/A	N/A	IVP, CTU	Incomplete	Left	Upper pole renal pelvis	Radical NU	N/A	Sarcomatoid TCC	Recurrence within 10 months Overall survival after recurrence 4 months
21	Unsal et al. [21]	2003	Turkey	68	F	Hematuria and right flank pain	S/P TAH + BSO due to endometrial carcinoma	Within normal limits	IVU, RPG, cystography (confirmed right side reflux) MRI	Complete	Right	Upper 1/3 of right upper moiety ureter	Hemi-NU+ bladder cuff	Laparotomy	TCC	N/A
22	Boris et al. [22]	2006	US	51	F	Hematuria	N/A	N/A	CTU, RPG	Incomplete	Right	Distal right ureter progress proximally into bifurcation	Radical NU	Laparotomy	T1 LG papillary UC	2 years NED
23	G. M. Chen et al. [23]	2011	Japan	77	M	LUQ abdominal pain	No per anamnesis	N/A	Dipstick, no evidence of microhematuria CT, MRI	Complete	left	Upper pole moiety 16 cm mass	Radical NU + bladder cuff	N/A	Invasive sarcomatoid TCC	Full recovery on follow-up (adjuvant chemotherapy and radiotherapy were offered however patient refused)
24	Kao et al. [24]	2012	Taiwan	87	F	Hematuria, abdominal fullness	No per anamnesis	Within normal limits	CTU	Unclear	Left	duplicated left kidney with the normal excretory upper moiety along with the hydronephrosis caused by tumor infiltration involving upper ureter of the lower moiety	HG UC was confirmed via urine cytology, considering her age; patients received palliative radiotherapy	-	-	6 months overall survival from diagnosis
25	LIN et al. [25]	2012	Taiwan	82	F	Hematuria	N/A	N/A	RPG	Complete	right	Lower moiety renal pelvis	Hemi-NU+ bladder cuff	Laparoscopic+ open for bladder cuff (Gibson’s incision)	pT1 HG UC	Two years follow-up NED
26	Ogawa et al. [26]	2014	Japan	71	F	Hematuria	N/A	N/A	CT, RPG	Incomplete	left	Left upper renal pelvis with invasion to renal parenchyma	Radical NU	N/A	pT4N0Mx squamous cell carcinoma	N/A
27	Zhang et al. [27]	2015	China	65	M	Hematuria and right flank pain	Smoking and drinking history Lives in Japan	N/A	US, CTU, MRU	Bilateral complete duplex collecting system on cystoscopy: right ectopic ureter insertion to posterior urethra	Right	Upper moiety renal pelvis	Radical NU	Laparotomy	Poorly differentiated UC showing invasive growth	N/A
28	Karray et al. [28]	2019	Tunisia	52	M	Hematuria	Smoking	Within normal limits	CTU	Complete + ectopic ureter	Right	Right upper pole ureter at the L2-4 level	Hemi-NU	Laparotomy	pT2G2 TCC	2 years follow-up NED
29	Sarkar et al. [29]	2020	India	46	M	Hematuria	No per anamnesis	N/A	CTU	incomplete	Right	Renal pelvis of lower moiety	Radical NU	Laparoscopic	Infiltrating TCC	N/A
30	Brnić, Zoran et al. [30]	2020	Croatia	63	M	Hematuria	N/A	Within normal limits	CTU	Complete	Left	Distal part of upper moiety ureter was tortuous and largely dilated (megaureter), crossing lower moiety ureter at the level of iliac crest. Two centimeters caudally from that point, within megaureter, a slightly hyperdense 3 cm × 5 cm irregularly shaped tumor was identified	Radical NU	Laparotomy	HG invasive UC T1NxMxG3	2 years follow-up NED
31	Storey et al. [31]	2021	Australia	76	M	Hematuria	N/A	N/A	CTU, RPG	Complete with ectopic ureter of upper pole moiety which inserts into bladder neck	Right	Upper pole moiety distal ureter mass	N/A	N/A	N/A	N/A
32	Nirei et al. [32]	2021	Japan	76	M	Hematuria	No per anamnesis	1.04 mg/dL	CT, MRU	Complete + ectopic upper moiety ureter opens to prostate verumontanum	Right	Upper moiety renal pelvis per imaging on diagnostic URS—to the ectopic ureteral opening beside the verumontanum confirmed no lower ureteral malignancy	Radical NU, because there was no obvious tumor around the ectopic ureter, lower ureter was blinded, and the prostate was preserved	Laparoscopic	High-grade pTa UC	6 months follow-up cystoscopy to blinded ectopic ureter—no recurrence sign
33	Sarver et al. [33]	2022	United States	76	M	Hematuria	S/P irradiation therapy for prostate adenocarcinoma S/P BCG instillation therapy for bladder LG UC	N/A	CTU, diagnostic URS	Incomplete	Left	At the level of bifurcation	Laser ablation of UTUC following biopsy confirmed T1 LG UC	Minimally invasive endoscopic management	-	N/A
34	Case presentation	2023	Israel	69	F	Hematuria	No per anamnesis	0.8	CTU, diagnostic URS	Complete	Left	Upper moiety renal pelvis	* Upper pole Hemi-NU with distal common sheath ureterectomy, including bladder cuff, and reimplantation of the lower moiety ureter *	Robotic assisted laparoscopic	Mucinous HG UC pT3NXM0	Continued adjuvant therapy 9 months follow-up NED

APG, antegrade pyelography; CT, computed tomography; CTU, CT urography; F, female; HG, high grade; Hemi-NU hemi-nephroureterectomy; IVP, intravenous pyelography; IVU, intravenous pyelography urography; M, male; LG, low-grade; LUQ, left upper quadrant; MRI, magnetic resonance imaging; MRU, magnetic resonance urography; N/A, not applicable; NED, no evidence of disease; NU, nephroureterectomy; RPG, retrograde pyelography; TCC, transitional cell carcinoma; UC, urothelial carcinoma; URS, ureteroscopy; US, ultrasonography.

**Table 2 jpm-14-00158-t002:** Descriptive characteristics of all reported cases with UTUC in DCS.

Patients		
No.	34	
Bilateral tumor	2	
Bilateral DCS with bilateral tumor	1	
Unilateral DCS with bilateral tumor	1	
Overall UTUC in DCS	35 units	
Year of report		
Range	1981–2023	
Age, years		
Mean (SD)	65.8	(11.1)
Median (IQR)	67	(62–73.5)
Region, no. (%)		
Australia	1	(2.94)
China	1	(2.94)
Croatia	1	(2.94)
India	1	(2.94)
Israel	1	(2.94)
Japan	18	(52.94)
Malaysia	1	(2.94)
Taiwan	3	(8.82)
Tunisia	1	(2.94)
Turkey	1	(2.94)
United Kingdom	1	(2.94)
United States	4	(11.76)
Gender, no. (%)		
Male	21	(61.75)
Female	13	(38.25)
Presentation, no. (%)		
N/A	6	
Hematuria	19	
Flank pain	1	
Abdominal pain	1	
Hematuria + flank pain	3	
Hematuria + abdominal pain	1	
Non-specific	3	
Total cases with hematuria	23	(67.5%)
Exposure, no.		
N/A	17	
None	10	
Smoking	3	
Radio/chemotherapy	3	
Other	1	
Creatinine (mg/dL), no.		
N/A	23	
Normal	8	
Abnormal	3	
Diagnosis, no.		
N/A	5	
CTU alone	6	
CTU combined with IVU/RPG/US/MRI	14	
RPG alone	1	
RPG/URS combined with other modalities	14	
Anatomy, no. (%)		
N/A	1	
Complete	15	(36.42%)
Incomplete	18	(51.42%)
Blind ending bifid ureter	1	
Side, no. (%)		
Right	18	(51.42%)
Left	17	(48.58%)
Bilateral with DCS	1	
Location, no. (%)		
Pelvis, upper moiety	10	(28.5%)
Pelvis, lower moiety	4	(11.5%)
Proximal ureter, upper moiety	1	(2.85%)
Proximal ureter, lower moiety	2	(5.7%)
Bifurcation	8	(22.5%)
Distal ureter, upper moiety	4	(11.4%)
Distal ureter, lower moiety	3	(8.5%)
Distal ureter, common sheath	2	(5.7%)
Ectopic ureter	1	(2.85%)
Management, no. (%)		
N/A	6	(17.14%)
Radical NU	12	(34.28%)
Radical NU with bladder cuff/partial cystectomy	9	(25.71%)
Heminephroureterectomy	5	(14.28%)
Other	3	(8.57%)
Approach, no.		
N/A	13	
Open	16	
Laparoscopic	3	
Robotic	1	
Endoscopic	1	
Other	1	
Pathology, no.		
N/A	11	
Undefined grade TCC	7	
Low-grade UC	3	
High-grade UC	8	
Sarcomatoid type UC	2	
Mucinous type UC	1	
Squamous cell carcinoma	1	
Poorly differentiated	1	
Follow-up, no.		
N/A	17	
2 years: NED	5	
Up to 2 years: recurrence	3	
Up to 1 year: NED	2	
Up to 1 year: recurrence	2	
Up to 6 months: NED	5	

CTU, CT urography; DCS, duplex collecting system; IVU, intravenous urography; MRI, magnetic resonance imaging; N/A, not applicable; NED, no evidence of disease; NU, nephroureterectomy; RPG, retrograde pyelography; TCC, transitional cell carcinoma; UC, urothelial carcinoma; URS, ureteroscopy; US, ultrasonography; UTUC, upper tract urothelial carcinoma.

## Data Availability

Data are contained within the article.

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
