# Peer review of "Management of Upper Tract Urothelial Carcinoma in a Double Collecting System Kidney"

_jpm, 2024, doi:10.3390/jpm14020158_

Round 1
Reviewer 1 Report
Comments and Suggestions for Authors
The manuscript titled "Management of Upper Tract Urothelial Carcinoma in a Double Collecting System Kidney" presents a detailed investigation of upper tract urothelial carcinoma (UTUC) in duplex collecting systems (DCS), a relatively rare condition. The authors have conducted a retrospective review of published cases and added their clinical experience with a specific case. Here are some comments and suggestions for improvement:
Major:
1. Please ensure that the search strategy are clearly defined. Is this a systematic review? It would be more valuable if it is a systematic review.
2. It would be useful to have an overall summary table regarding patient demographics of all included studies, including age (median, IQR), gender (percentage), DCS type (percentage), UTUC location (percentage), treatment options (percentage) etc. It would be easier for readers to digest than only Table 1.
Minor:
1. Fig 3 needs to be higher quality and have clearer text.
2. Page 15, line 370 should be “UTUC”, not “UCUT”
3. Page 16, line 422 should be severe not sever.
Comments on the Quality of English LanguageGood quality.
Author Response
Point-to-point response to reviewer’s 1 comments
Dear referee,
Thank you for taking the time to review this manuscript. Your comments have helped as to strengthen our points. Please find the detailed responses below and the corresponding revisions/corrections highlighted via track changes in the re-submitted manuscript.
Major comments:
- The search strategy is mentioned first page 1, line 15, abstract “…conducted a systematic search”
To stratify this we added in page 2 line 70 “The search strategy was systematic and included all UC cases in DCS.”
- A descriptive table of overall cases was added and included 34 patients, out of which 2 patients had a bilateral presentation; thus, overall, 36 cases were included.
See page 3 line 106-107.
The table is referred to as Table 2 on pages 13-14 following ‘4. Results’
Minor comments:
- Fig 3 needs to be higher quality and have clearer text –
The changes have been made on page 11 of the main manuscript. Figure 3 has been completely changed, and a short explanation has been added to every picture.
- Page 15, line 370 should be “UTUC”, not “UCUT” – change was made in page 16 line 381
- Page 16, line 422 should be severe not sever – change was made page 17 line 433
Reviewer 2 Report
Comments and Suggestions for Authors
The topic is worthy of interest, and few literature data are currently available.
A full revision of the scientific English language should be desirable, whether possible with the help of a mother tongue expert (see the section below).
Some other comments:
Line 32 Introduction: whether the aim of the presented paper is to focus on the management of such a rare combination urological malignancy on abnormal collecting systems, the introduction should start with information on UTUC, and subsequent general description of DCS.
Line 49 “Urothelial carcinoma (UC) is the 4th most common solid organ malignancy: worldwide? GLOBOCAN data should be mentioned in addition to the reported reference.
Line 54 “A historical population-based cohort study followed army recruits…”: did the authors mean followed-up?
Line 69: “Research strategy” or “Data collection strategy” may be more appropriate.
Line 82: “out of urothelial origin” is the correct form
Line 100 This paragraph is mostly incorrect and needs to be re-written as follows: “The selected cases were manually analyzed, and the exstracted data were summarized in a table according to the available information. Subsequently, an original case report was added, based on our clinical experience”
Line 106: “The obtained data are presented in Table 1” is a more appropriate form.
Line 202 “Six weeks after surgery” is more appropriate.. as well as “the patient presented a good PS..” and “Hystopathological report (remove “A”) confirmed…”
Line 209 “Disease staging based on pathology report concluded pT3NxM0 urothelial carcinoma”. Moreover, the TNM staging reference should be added.
Line 211 “The case was presented at our Institutional multidisciplinary tumor board” may be more appropriate.
Line 214 “…which ruled out distant metastasis”: did the authors mean “to exclude distant metaatases”?
Line 224: “She was followed up every 3 months.”. What about the living state at the last follow up?
Line 227 “Of these, 2 cases were bilateral”: whether 2 patients showed bilateral cancer should be better specified.
Line 248: “Table 1 presents the descriptive data.” There is too much repetition of the word “data” in a single sentence. Population characteristics or “the characteristics of the population in study” may be more appropriate.
Line 255-257 “few of these surgeries were aimed at diagnosis rather than definitive treatment, at times, diagnostic ureteroscopy was unavailable. In addition, these surgeries…”: this sentence is not clear and should be reformulated. Are the authors referring to different surgical techniques?
Line 394: The need for postoperative, adjuvant treatment has been mentioned in the Conclusions, the authors also briefly described their experience with multidisciplinary tumor board discussion to evaluate this option after surgery, but this topic is lacking in the Discussion. The authors should broaden the section “oncological considerations” (or add a dedicated paragraph after the wide surgical description) with information of the role/the need for adjuvant or neoadjuvant treatment and/or multimodal therapeutic strategies (Radiotherapy? Chemo-/target therapy?) in the presented setting.
Comments on the Quality of English LanguageA full revision of the scientific English language should be desirable, whether possible with the help of a mother tongue expert (i.e. Abstract: “The patient underwent robotic…” ; “To conclude, UTUC of DCS is rare and underreported (underreported is redundant, a more technical and effective synonym is desirable), thus it is difficult to define a standard treatment….” ; “Despite hemi-nephroureterectomy has already been described in literature, ….” ; “…to the best of our knowledge, this is the first report of robot-assisted hemi-nephroureterectomy for complete DCS…”; Line 251 “…all the available published data…” ; Line 271 “technological advances” ; and so on).
Author Response
Point-to-point response to reviewer’s 2 comments
Dear referee,
Thank you for taking the time to review this manuscript. Your comments have helped as to strengthen our points. Please find the detailed responses below and the corresponding revisions/corrections highlighted via track changes in the re-submitted manuscript.
- Comments and Suggestions for Authors:
- Line 32 Introduction: whether the aim of the presented paper is to focus on the management of such a rare combination urological malignancy on abnormal collecting systems, the introduction should start with information on UTUC, and subsequent general description of DCS.
We moved line number 53 which discusses UTUC to the beginning of the introduction to refocus the article point of interest.
See page 1 line 32 “Urothelial carcinoma (UC) is the 4th most common solid organ malignancy; however, only 5-10% arises from the upper urinary tract [1]. Thus, the prevalence of primary upper tract urothelial carcinoma (UTUC) in DCS is underreported with an unknown estimation rate.”
- Line 49 “Urothelial carcinoma (UC) is the 4th most common solid organ malignancy: worldwide? GLOBOCAN data should be mentioned in addition to the reported reference.
Line 49: currently line 32, an additional reference was added to strengthen the mentioned data-
[2] Jubber I, Ong S, Bukavina L, et al. Epidemiology of Bladder Cancer in 2023: A Systematic Review of Risk Factors. Eur Urol. 2023.
The data is based on the GLONOCAN data
The following comments where changed after full revision of the paper with the help of mother tongue expert for English as requested:
- Line 54 “A historical population-based cohort study followed army recruits…”: did the authors mean followed-up?
- Line 69: “Research strategy” or “Data collection strategy” may be more appropriate.
- Line 82: “out of urothelial origin” is the correct form
- Line 100 This paragraph is mostly incorrect and needs to be re-written as follows: “The selected cases were manually analyzed, and the exstracted data were summarized in a table according to the available information. Subsequently, an original case report was added, based on our clinical experience”
- Line 106: “The obtained data are presented in Table 1” is a more appropriate form.
- Line 202 “Six weeks after surgery” is more appropriate. as well as “the patient presented a good PS..” and “Hystopathological report (remove “A”) confirmed…”
- Line 209 “Disease staging based on pathology report concluded pT3NxM0 urothelial carcinoma”. Moreover, the TNM staging reference should be added.
[34] Brierley J, Gospodarowicz MD, Wittekind CT. TNM Classification of Malignant Tumors International Union Against Cancer. 8th. Oxford, England: Wiley; 2017. Wiley. 2017;
- Line 211 “The case was presented at our Institutional multidisciplinary tumor board” may be more appropriate.
- Line 214 “…which ruled out distant metastasis”: did the authors mean “to exclude distant metastases”?\
- Line 224: “She was followed up every 3 months.”. What about the living state at the last follow up?
248: “Subsequently, she underwent regular follow-ups every three months with a urologist, concurrently maintaining active oncology surveillance. It is noteworthy that during this period, the patient reported fatigue and nausea, which subsequently resolved upon the completion of adjuvant chemotherapy (ACT). Her current physical status is unremarkable, with a score of 0 in the ECOG score.”
- Line 227 “Of these, 2 cases were bilateral”: whether 2 patients showed bilateral cancer should be better specified.
257: “Among these, two cases were bilateral; one presented bilateral DCS with bilateral UTUC, while the second case exhibited bilateral tumours, with only one side having the anatomical variation of DCS (Table 1: patients 17 and 13, respectively)”
- Line 248: “Table 1 presents the descriptive data.” There is too much repetition of the word “data” in a single sentence. Population characteristics or “the characteristics of the population in study” may be more appropriate.
- Line 255-257 “few of these surgeries were aimed at diagnosis rather than definitive treatment, at times, diagnostic ureteroscopy was unavailable. In addition, these surgeries…”: this sentence is not clear and should be reformulated. Are the authors referring to different surgical techniques?
- Line 394: The need for postoperative, adjuvant treatment has been mentioned in the Conclusions, the authors also briefly described their experience with multidisciplinary tumor board discussion to evaluate this option after surgery, but this topic is lacking in the Discussion. The authors should broaden the section “oncological considerations” (or add a dedicated paragraph after the wide surgical description) with information of the role/the need for adjuvant or neoadjuvant treatment and/or multimodal therapeutic strategies (Radiotherapy? Chemo-/target therapy?) in the presented setting.
- 4. Oncological considerations
- The oncological management of UTUC is intricate, aligning with the patient's overall medical status, expectations, and tumor features to ensure positive outcomes. For low-risk/LG disease, a renal-sparing procedures are recommended. However, the management of HG/high-risk UC is more restrictive and requires carful multidisciplinary planning[34].
- Although conclusive results have not been published, neoadjuvant chemotherapy (NAC) may be considered for patients with advanced disease, or where post-operative renal function precludes ACT. Studies have shown that NAC reduces disease recurrence and mortality rates. Furthermore, some research reports pathological downstaging in case of metastatic disease, leading to favorable operative outcomes[57,58]. The usage of immunotherapy as neoadjuvant for cisplatin unfit patients was assessed in a small phase II randomized control trial (RCT), and did not prove to be advantageous [59].
- After (NAC), surgery is contemplated. According to the EAU guidelines, kidney-sparing surgery for UTUC should be considered in cases of a single kidney or impaired renal function based on specific cases. In this instance, we were not obligated to preserve renal function since the patient’s GFR was >60 ml/min and she did not exhibit any medical comorbidities that could cause a decrease in renal function [34]. However, based on the Nature review from 2015 – nowadays, thanks to precise surgical tools and sensitive radiological examinations, nephron-sparing procedures can be considered even in healthy young adults, if close clinical follow-up can be maintained[49]. Open radical NU (RNU) with bladder cuff excision is the gold standard therapy for HG/high risk UTUC[34]. The minimally invasive approach is acceptable when oncological principles are maintained. Current data confirms that the robotic approach is superior to open surgery in terms of postoperative complications and not inferior in terms of oncological outcomes, thus increasing its popularity[60].
- The primary concern with renal sparing surgical therapy, such as hemi-NU, for UTUC lies in the oncological outcomes. Firstly, considering the relatively high rate of seeding contamination for UC, minimizing intraoperative contamination is crucial[53]. According to the revised and updated 2023 EAU guidelines, for UTUC, diagnostic URS can be employed, preferably without a biopsy, only when imaging and/or voided urine cytology prove insufficient for the diagnosis and/or risk stratification of patients suspected to have UTUC[34].
- Our analysis reveals that 15 out of 34 cases (44%) were diagnosed with a UC based on retrograde pyelography (RPG) or diagnostic URS. In DCS, especially in incomplete anatomy, the significance of intraluminal contamination becomes crucial when planning hemi-NU. However, the underreported follow-up information in our table complicates the estimation of the rate of seeding contamination for DCS with UTUC diagnosed with RPG/URS. Nevertheless, our patient, who underwent diagnostic ureteroscopy (URS) with biopsy prior to radical resection, was reassessed by cysto-ureteroscopy and showed no evidence of recurrence. Still, in alignment with updated guidelines, consideration should be given to minimizing the use of these interventional diagnostic tools when non-interventional imaging is feasible[34].
- When conducting radical resection, bladder cuff excision is imperative, significantly reducing the recurrence rate[61]. In contrast, intraoperative nodal management is not obligatory. In 30%-40% of radical NU surgeries, a suspected lymph node is visible. However, the benefits of lymph node dissection (LND) for node-negative (cN0) UTUC have not yet been established[62]. Notably, while LND improves pathological staging; the updated EAU guidelines do not recommend LND for cT1N0 disease [34]. Consequently, we did not perform LND in our patient. For cT2-T4, a recent multi-institutional study concluded that LND did not improve survival in patients with cN0 disease. In positive-node disease, prognosis is directly related to the number of positive nodes removed and the dissection template[62]. It is essential to align the primary tumor location to the nodal dissection template because ureteral tumors are locally more advanced and typically metastasize earlier than renal pelvic tumors [63].
- Platinum-based ACT is recommended for pT2-T4/any N/ M0 disease within 90 days of surgery to prolong disease free survival (DFS)[64]. For cisplatin ineligible patients, carboplatin can serve as an alternative[65]. Immunotherapy, particularly checkpoint inhibitors (CIs), may also be considered. These ICs are frequently used as an alternative treatment for platinum-resistant disease or as maintenance therapy after cisplatin/carboplatin + gemcitabine adjuvant chemotherapy in cases of non-progressive disease[66–71]. Current cancer research is focusing on immune-phenotype-based therapy to enhance tumor response[72]. Radiotherapy is less commonly employed, with limited information available regarding its efficacy as a combination therapy. Typically, it is used for targeted lesions[73,74].
- Comments on the Quality of English Language
A full revision of the scientific English language should be desirable, whether possible with the help of a mother tongue expert (i.e. Abstract: “The patient underwent robotic…” ; “To conclude, UTUC of DCS is rare and underreported (underreported is redundant, a more technical and effective synonym is desirable), thus it is difficult to define a standard treatment….” ; “Despite hemi-nephroureterectomy has already been described in literature, ….” ; “…to the best of our knowledge, this is the first report of robot-assisted hemi-nephroureterectomy for complete DCS…”; Line 251 “…all the available published data…” ; Line 271 “technological advances” ; and so on).
Full revision has been made by the institutional English speakers’ advisors. Changes are noted in the main manuscript Word sheath and highlighted by the author.
Round 2
Reviewer 2 Report
Comments and Suggestions for Authors
I endorse the publication of the revised paper.
Just one more little editing suggestion:
Line 126, Line 132 and Line 146: Figures should be uniformly mentioned as reported in the relative captions, that is, “Figure 1/2/3” or “Fig. 1/2/3” in both the main text and Figure legends.
Author Response
Dear reviewer,
Thank you for taking the time to review this manuscript. We appreciate your comments and changed the abbreviation 'Fig.' to the word Figure in the corresponding lines.